# Evolutionary Events Promoted Polymerase Activity of H13N8 Avian Influenza Virus

**DOI:** 10.3390/v16030329

**Published:** 2024-02-21

**Authors:** Bo Meng, Qian Wang, Haoyu Leng, Chenyang Ren, Chong Feng, Weiwei Guo, Yali Feng, Ying Zhang

**Affiliations:** Key Laboratory of Livestock Infectious Diseases, Ministry of Education, Liaoning Key Laboratory of Zoonosis, Laboratory of Ruminant Infectious Disease Prevention and Control (East), Ministry of Agriculture and Rural Affairs, Liaoning Panjin Wetland Ecosystem National Observation and Research Station, College of Animal Science and Veterinary Medicine, Shenyang Agricultural University, 120 Dongling Rd., Shenyang 110866, China2022200173@stu.syau.edu.cn (C.R.);

**Keywords:** wild bird, avian influenza virus, H13N8, H9N2, evolution, ribonucleoprotein complexes

## Abstract

Wild birds are considered to be the natural reservoir hosts of avian influenza viruses (AIVs). Wild bird-origin AIVs may spill over into new hosts and overcome species barriers after evolutionary adaptation. H13N8 AIVs used to be considered primarily circulated in multispecies gulls but have recently been shown to possess cross-species infectivity. In this study, we analyzed the genetic changes that occurred in the process of the evolution of H13 AIVs. Phylogenetic analysis revealed that H13 AIVs underwent complex reassortment events. Based on the full genomic diversity, we divided H13 AIVs into 81 genotypes. Reassortment experiments indicated that basic polymerase 2 (PB2) and nucleoprotein (NP) genes of the H9N2 AIV significantly enhanced the polymerase activity of the H13N8 AIV. Using the replication-incompetent virus screening system, we identified two mutations, PB2-I76T and PB2-I559T, which could enhance the polymerase activity of the H13N8 AIV in mammalian cells. Notably, these mutations had been acquired by circulating H13N8 AIVs in 2015. These findings suggest that H13N8 AIVs are about to cross the host barrier. Occasional genetic reassortments with other AIVs and natural mutation events could promote this process. It is imperative to intensify monitoring efforts for H13N8 AIVs.

## 1. Introduction

Avian influenza viruses (AIVs), classified within the *Orthomyxoviridae* family, possess genomes consisting of eight negative-sense, single-stranded RNA segments. According to the two surface glycoproteins, hemagglutinin (HA) and neuraminidase (NA), AIVs are classified into 16 HA (H1–H16) and 9 NA (N1–N9) subtypes [1]. Wild birds serve as the natural reservoir hosts of AIVs, and technically, all AIV subtypes can be isolated from them. Due to their migration habits, wild birds have the capacity to carry AIVs over wide geographic distances [2]. From natural hosts, whole viruses or partial gene segments of wild bird-origin AIVs occasionally jump to new hosts, including domestic poultry and mammals, and may further form new lineage circulated in new hosts.

H13 subtype AIVs exhibit a narrower host range, primarily infecting gulls [3]. Gulls can excrete H13 AIVs through their oropharynx and cloaca. H13 AIVs were first isolated in 1977 from gulls in the United States [4]. Since then, H13 AIVs have been detected in various regions, including North America, South America, Europe, Asia, Africa, and Oceania [5]. At present, H13 AIVs have been diversified into Eurasian and North American lineages. Intercontinental and interspecies genome reassortment events frequently occurred between these two lineages because of the bird migrations [5,6,7]. Previous studies indicated that domestic poultry was resistant or refractory to infection with H13 AIVs. However, recent studies reported that the H13 viruses have been spread from migratory birds into domestic poultry, as H13-specific seroconversion has been detected in farm chickens situated along the migratory routes of black-tailed gulls [7]. In addition to that, H13N8 AIVs have been found to replicate efficiently in young chickens and mice [8]. These observations suggest that H13N8 AIVs possess cross-species infectivity, but the related mechanisms have not been fully elicited.

Since first identified in United States turkeys in 1966, H9N2 AIVs have spread widely around the world [9,10,11]. H9N2 AIVs are the most prevalent subtypes in China and are endemic in multiple avian species [12,13]. Meanwhile, H9N2 AIVs have acquired the ability to overcome species barriers, resulting in infections in pigs, dogs, and even humans [14]. More importantly, the internal gene cassette of H9N2 AIVs contributes to the higher adaptation to the new hosts of emerging reassortment viruses, such as H3N8, H5N6, H7N9, and H10N8 AIVs.

α2,6-linked sialic acid (human-like) receptor binding ability of HA and the viral RNA polymerase activity are considered key factors affecting the cross-species infection ability of AIVs. It has been confirmed that the HA protein of H13 AIVs tends to bind to the α2,3-linked sialic acid (avian-like) receptor [8]. Therefore, in this study, we focused on the RNA polymerase activity of the H13N8 AIV to identify mutations related to cross-species infection ability in polymerase-related genes. We first investigated the evolutionary characteristics of H13 AIVs. Then, we swapped the polymerase-related genes of the H13N8 AIV with those of the H9N2 AIV to assess the potential impact of reassortment events. Additionally, we employed the replication-incompetent, enhanced green fluorescence protein (EGFP)-expressing virus system to screen for mutations that could enhance the polymerase activity of the H13N8 AIV. These findings will contribute to a better understanding of the mammalian adaptation process of H13N8 AIVs and enable us to be aware of the threatening strains in the future.

## 2. Materials and Methods

### 2.1. Cells and Viruses

Human embryonic kidney 293T cells (HEK293T) were cultured in Dulbecco’s modified Eagle’s medium (DMEM; Gibco, Waltham, MA, USA) containing 1% penicillin–streptomycin and 10% fetal bovine serum (FBS; Gibco, Waltham, MA, USA). Madin-Darby canine kidney (MDCK) cells were cultured in DMEM supplemented with 5% FBS. MDCK cells stably expressing the HA protein (MDCK-HA) were gifted by Zoonotic and Exotic Infection Diseases Division, Harbin Veterinary Research Institute, Chinese Academy of Agricultural Sciences and cultured in DMEM supplemented with 5% FBS containing 800 µg/mL Geneticin (Gibco, Waltham, MA, USA). The cells were incubated at 37 °C in a humidified incubator with 5% CO_2_.

Two AIV strains used in this study, A/Eurasian curlew/Liaoning/ZH-385/2014(H13N8) (ZH385; Accession Number: KR010440-KR010447) [8] and A/chicken/China/07/2016(H9N2) (CKLN07; Accession Number: MW255942-MW255949) [15] were isolated from Eurasian curlew and chicken in Liaoning in 2014 and 2016. All viruses were propagated in embryonated chicken eggs at 37 °C and stored at −80 °C.

### 2.2. Phylogenetic Analysis

For phylogenetic analysis of H13 AIVs, reference sequences with the complete coding regions of 550 HA, 854 NA, 854 PB2, 854 basic polymerase 1 (PB1), 953 acidic polymerase (PA), 854 NP, 854 matrix (M), and 854 nonstructural (NS) genes were downloaded from the Bacterial and Viral Bioinformatics Resource Center (https://www.bv-brc.org/; (accessed on 7 December 2023)). Details of reference sequences are summarized in Appendix A. Sequence alignment was conducted with Maft v7.490 [16]. The phylogenic tree was built with the maximum-likelihood method with rapid bootstrap replicates using Fast Tree v2.1.11 [17].

### 2.3. Dual-Luciferase Reporter Assay

Polymerase activity analysis was performed by using the dual-luciferase reporter assay system as previously described [18,19]. Briefly, the polymerase-related genes, PB2, PB1, PA, and NP, of ZH385 and CKLN07, were cloned into the mammalian expression vector pCAGGS. According to the fluorescence-activated cell sorting results, mutations in the PB2 protein were introduced into ZH385 PB2 plasmids, respectively, by site-directed mutagenesis PCR. For the reporter, an influenza minigene containing luciferase pPolI-NP-luci plasmid was used. For standardization, a Renilla luciferase-expressing plasmid pRL-TK was used. The pPolI-NP-Luci plasmid could express a virus-like RNA carrying the complete ORF of the firefly luciferase gene under the control of the human RNA polymerase I promoter. HEK293T cells were transfected with pPolI-NP-Luci plasmid, pRL-TK plasmid, and PB2, PB1, PA, and NP expression plasmids. Transfected cells were incubated for 24 h at 37 °C. The luciferase activity was assayed by using the Luciferase assay kit (Vazyme, Nanjing, China). All experiments were performed in triplicate.

### 2.4. Generation of Influenza Viruses by Reverse Genetics

The influenza viruses were generated by using the reverse genetics system as described previously [20]. Briefly, we inserted the eight gene segments of the ZH385 virus into the vRNA-mRNA bidirectional transcription vector pBD with a Clone Express II One Step Cloning Kit (Vazyme, Nanjing, China).

In the generation of the replication-incompetent ZH385, the modified HA-EGFP gene segment, which encoded a virus-like RNA of ZH385 containing the 3′ HA noncoding regions, 48 nucleotides corresponding to the HA coding sequence at the 3′ end of the vRNA, the EGFP coding sequence, 291 nucleotides corresponding to the HA coding sequence at the 5′ end of the vRNA, and the 5′ HA noncoding region, was inserted into the vector pBD as described previously [21]. We replaced the wild-type plasmid for HA with the corresponding modified HA-EGFP plasmid. The pBD-PB2, PB1, PA, NP, NA, M, NS, and HA-EGFP plasmids were co-transfected into HEK293T cells to generate a replication-incompetent virus. Moreover, cells were transfected with the eukaryotic protein expression plasmid encoding the HA protein of ZH385. The replication-incompetent ZH385 was propagated in MDCK-HA cells. The replication of the replication-incompetent ZH385 virus was restricted to a cell line that stably expresses the HA protein (Appendix A).

In the generation of the replication-incompetent virus libraries, random mutations were introduced into the PB2 and NP genes of ZH385 by error-prone PCR using Gene Morph II Random Mutagenesis Kit (Agilent, Santa Clara, CA, USA) as described previously [21,22]. Briefly, PCR reaction conditions and target DNA template amounts were optimized to generate 1–2 amino acid mutations per gene. The randomly mutated PCR products were then inserted into the pBD vector to generate randomly mutated plasmid libraries. The replication-incompetent virus libraries were generated similarly, but we replaced the wild-type PB2 or NP plasmid with the corresponding randomly mutated plasmid library.

We also generated the ZH385 virus as the control. According to the polymerase activity results, three mutants possessing PB2-I76T, PB2-I559T, and PB2-I76T/I559T mutations were also generated, respectively, and propagated in MDCK cells.

### 2.5. Fluorescence-Activated Cell Sorting by FACS

A total of 10^3.5^ 50% tissue culture infectious doses (TCID_50_) of the replication-incompetent ZH385 virus and the replication-incompetent virus libraries were introduced into MDCK cells in 100 µL respectively. At 6 h post-infection, FACS was carried out using FACS Aria III (BD Biosciences, Franklin Lakes, NJ, USA). The cell sorting gate was set to capture the cells infected with the replication-incompetent virus libraries that expressed higher EGFP intensity than the wild-type ZH385. To propagate the viruses with increased EGFP expression levels, screened-out single cells were sorted into 96-well plates covered with confluent monolayer MDCK-HA cells and incubated at 37 °C. At 72 h post-infection, the EGFP expression levels in the sorted cell wells were measured by using Cytation 5 (Bio Tek, Winooski, VT, USA). The cell cultures with the EGFP expression were stored at −80 °C for sequencing.

### 2.6. Quantification of Viral RNA Species

MDCK cells were seeded in 12-well plates and infected with 10^3.5^ TCID_50_ of ZH385, PB2-I76T, PB2-I559T, and PB2-I76T/I559T viruses. At 6 h and 8 h post-infection, the total RNA was extracted using a Trizol reagent (Magen, Guangzhou, China). Relative quantities of vRNA, cRNA, and mRNA were determined by qRT-PCR as previously described [23]. The primer sequences used for the reverse transcription were AGCAAAAGCAGG (vRNA), 5′-CCTTGTTTCTACT-3′ (cRNA), and TTTTTTTTTTTTTTTTTT (mRNA). Subsequently, for qRT-PCR, the primer sequences 5′-GGGGTTGGGACAATGGTGAT-3′ (forward) and 5′-CGTTGTGCTGCTGTTTGGAA-3′ (reverse) were used to amplify the ZH385 NP gene, and the primer sequences 5′-GGTCACCAGGGCTGCTTTTA-3′ (forward) and 5′-TCCCGTTGATGACAAGTTTC-3′ (reverse) were used to amplify the GAPDH gene. Expression values for each gene relative to GAPDH were calculated by using the 2^−ΔΔCt^ method. All experiments were performed in triplicate.

### 2.7. Statistical Analysis

Statistical analysis between different groups was performed by one-way analysis of variance (ANOVA) test using Graph Pad Prism version 8.0 (Graph Pad Software Inc., San Diego, CA, USA). The difference with a value of *p* < 0.05 was considered statistically significant, while *p* < 0.01 was considered highly statistically significant.

## 3. Results

### 3.1. Phylogenetic Analysis of H13 AIVs

We systematically analyzed the evolutionary characteristics of H13 subtype AIVs. The phylogenic tree results showed that, apart from the NA genes, which formed five branches, the genome segments of H13 AIVs clustered into three distinct branches. It seemed that the HA, NA, PB1, and NP gene of ZH385 were derived from a NL/1/00(H13N8)-like virus, whereas PB2, PA, M, and NS genes originated from SE/1/2005(H13N8)-like, AK/44199-097/2006(H13N3)-like, SB/272/1998(H13N6)-like, and DE/520/1988(H13N9)-like viruses, respectively (Figure 1 and Appendix A).

On the basis of the full genomic diversity, the 550 H13 AIVs were divided into 81 different genotypes (Appendix A). Among the 81 genotypes, 12 genotypes, including genotype 5 (G5), 6 (G6), 33 (G33), 47 (G47), 64 (G64), 67 (G67), 68 (G68), 69 (G69), 72 (G72), 75 (G75), 77 (G77), and 79 (G79), possessed more than ten strains. According to the geographical distribution of these 12 genotypes, the G5, G6, G33, and G47 viruses were mainly prevalent in North America and South America, whereas the G64, G67, G68, G69, G72, G75, G77, and G79 viruses were prevalent in Europe and Asia (Figure 2). Since 2015, the G5, G6, G33, G47, and G67 genotypes have become more prevalent, while the ZH385 virus belonged to G68. The above results demonstrated that the circulating H13 AIVs had undergone complex genomic reassortment events.

### 3.2. PB2 and NP Genes of CKLN07 Significantly Increases the Polymerase Activity of ZH385

Genome reassortment events have led to the rapid evolution of AIVs, assisting the viral cross-species infection ability. H9N2 viruses are now considered to be the common internal gene cassette provider of emerging zoonotic AIVs. To investigate whether recombination with H9N2 AIVs could enhance the polymerase activities of H13N8 AIVs, we swapped the PB2, PB1, PA, and NP genes of ZH385 with CKLN07. The results showed that the polymerase activity of the CKLN07 virus was 2.96 times greater than that of ZH385 (Figure 3A). However, compared to the ZH385 polymerase, the reassortment polymerase containing the PB2 or NP gene of the CKLN07 virus increased the polymerase activity by 4.22-fold and 2.39-fold, respectively. When the PB2 and NP genes of the ZH385 virus were together replaced with the CKLN07 virus, the polymerase activity was increased by 7.42-fold compared to ZH385. The above results indicated that the PB2 and NP genes were restrictive genes for the polymerase activity of the H13N8 virus in HEK293T cells. Moreover, reassortment of polymerase-related genes between H13N8 and H9N2 viruses might significantly affect the polymerase activity.

### 3.3. Identify Mutations That Enhanced Polymerase Activity of ZH385

MDCK-HA cells were infected with 10^3.5^ TCID_50_ of the replication-incompetent ZH385, the PB2 replication-incompetent virus library, and the NP replication-incompetent virus library, respectively. After 6 h post-infection, we performed FACS analysis on the infected cells. In the PB2 replication-incompetent virus library, we screened out 8636 expressing EGFP cells and 137 cells with increased EGFP expression levels compared with the replication-incompetent ZH385 (Appendix A). In the NP replication-incompetent virus library, 8219 cells were screened, but only two cells had increased EGFP expression levels (Appendix A). To propagate the viruses with increased EGFP expression levels, the selected cells were then individually sorted into a 96-well plate covered with MDCK-HA cells. After 72 h, the EGFP expression levels in the sorted cell wells were measured. The EGFP expression was detected in 41 sorted cell wells of the PB2 replication-incompetent virus library, whereas no EGFP expression was detected in sorted cell wells of the NP replication-incompetent virus library. The EGFP expression cells were harvested and sequenced. Six samples possessed PB2 mutations, including PB2-D60E, PB2-I76T, PB2-I382V, PB2-I461V, PB2-I559T, and PB2-N711D.

### 3.4. Screened Mutations in PB2 Protein Enhanced the Polymerase Activity of ZH385

We then introduced PB2-D60E, PB2-I76T, PB2-I382V, PB2-I461V, PB2-I559T, and PB2-N711D mutations into the PB2 expressing plasmid, respectively, and evaluated the effect of these mutations on the polymerase activity of ZH385 in HEK293T cells. As shown in Figure 3B, the PB2-I76T and PB2-I559T mutations enhanced the polymerase activity of ZH385 by 1.69-fold and 2.72-fold, respectively. The remaining four mutations in PB2 did not affect the polymerase activity of ZH385 dramatically. We further investigated the combined effect of PB2-I76T and PB2-I559T mutations. As shown in Figure 3C, the PB2-I76T/I559T mutation enhanced the polymerase activity of ZH385 by 2.70-fold. However, no additive effects were observed in the PB2-I76T/I559T mutation compared with the PB2-I559T mutation. The above results indicated that the PB2-I76T and PB2-I559T mutations significantly increased the polymerase activity of H13N8 AIVs in mammalian cells.

### 3.5. The Screened PB2 Mutations Enhanced the Viral RNA Replication and Transcription

During the process of influenza virus replication and transcription, two different classes of RNA, complementary RNA (cRNA) and messenger RNA (mRNA), are synthesized from the viral RNA (vRNA) templates. The cRNA is used as the template for vRNA replication, whereas the mRNA serves as the template for guiding the synthesis of viral proteins. To investigate the effect of PB2-I76T, PB2-I559T, and PB2-I76T/I559T mutations on the replication and transcription of ZH385, the cRNA, vRNA, and mRNA levels were quantified using qRT-PCR. We found that the vRNA and mRNA levels were significantly increased in MDCK cells infected with the PB2-I76T, PB2-I559T, and PB2-I76T/I559T mutants (Figure 4A,B,E,F). The cRNA level in the cells infected with the PB2-I559T and PB2-I76T/I559T mutants were significantly increased compared to ZH385, whereas no significant difference was detected in the cRNA level between the PB2-I76T mutant and ZH385 (Figure 4C,D). These results demonstrated that the PB2-I559T and PB2-I76T/I559T mutations both enhanced the replication and transcription in the life cycle of H13N8 AIVs, while the PB2-I76T mutation primarily affected mRNA transcription.

### 3.6. PB2-I76T and PB2-I559T Mutations Were Prevalent in H13N8 AIVs Isolated in Recent Years

To investigate the prevalence of mutations PB2-I76T and PB2-I559T, we summarized the amino acids at positions 76 and 559 in PB2 proteins of H13N8 AIVs. Among 139 H13N8 AIVs, 102 possessed PB2-76I, 37 possessed PB2-76T; 99 possessed PB2-559V, 37 possessed PB2-559T, two possessed PB2-559A, and one possessed PB2-559I (Figure 5 and Appendix A). PB2-76I was detected in all strains isolated from 2000 to 2014, while PB2-76T was detected in strains isolated after 2015. From 2000 to 2014, PB2-559V was dominant, while PB2-559A and PB2-559I were only detected in a few strains. PB2-559T was detected in all strains isolated since 2015.

## 4. Discussion

H13 AIVs have been widely detected among wild birds across numerous countries for the last few decades. Based on the geographic distribution, H13 AIVs are divided into Eurasian and North American lineages [24]. In this study, we analyzed the genomic evolutionary characteristics of all available H13 isolates. The eight gene segments of the Liaoning isolates, ZH385, belonged to Eurasian lineages but originated from different subtype viruses. Genomic analysis revealed that on the basis of the full genomic diversity, H13 AIVs formed 81 distinct genotypes through reassortment among themselves or with H16 AIVs circulating in wild birds. Previous studies have suggested that intercontinental reassortment events were frequently found among H13 and H16 AIVs [6,25]. Among the 81 genotypes, 12 genotypes possessed more than ten strains. We found that the 12 genotypes exhibited certain geographic and date differences. The G5, G6, G33, and G47 viruses were mainly prevalent in North America and South America, whereas the G64, G67, G68, G69, G72, G75, G77, and G79 viruses were prevalent in Europe and Asia. In addition to that, the G5, G6, G33, G47, and G67 genotypes have been prevalent since 2015. The results showed that the evolution events continuously occurred during H13 AIVs circulating in nature.

Reassortment events enable AIVs to acquire adaptation features for the new hosts. The 1918, 1957, and 2009 human influenza pandemic strains were all generated by viral reassortment events. It had been reported that the reassortment between the H5N1 and 2009 pandemic could extensively enhance the pathogenicity and transmissibility of H5N1 AIVs in mammalian hosts [26]. H9N2 AIVs actively participate in gene reassortment and are the common internal gene cassette provider of emerging zoonotic AIVs. The PB1 gene of H9N2 AIVs could increase the polymerase activity of H1N1 AIVs. H9N2-derived PB2 and PA genes increased polymerase activity in the dominant G72 genotype H7N9 virus in human cells [27]. In this study, we found that the recombination with H9N2 AIV PB2 and NP genes significantly increased the polymerase activity of the H13N8 AIV. The results indicated that reassortment events between H13N8 and H9N2 AIVs could enhance the polymerase activities of H13N8 AIVs. The expression levels of PB2 and NP proteins of H13N8 and H9N2 AIVs were not measured in this study. There was a possibility that PB2 and NP proteins of H9N2 AIVs might potentially exhibit higher expression levels compared to those of H13N8 AIVs, which might increase the polymerase activities.

To identify mutations that enhance the polymerase activities of H13N8 AIVs, we employed a high-throughput screening system. Random mutations were introduced into the PB2 and NP genes of H13N8 AIV, ZH385. Then, the PB2 and NP replication-incompetent virus libraries expressing EGFP protein were generated. The wild type of HA gene was replaced with the modified HA-EGFP gene. Under this circumstance, the replication of the replication-incompetent virus libraries was restricted to the HA protein-expressing cell line. We sorted the fluorescence-activated cells infected with the PB2 or NP replication-incompetent virus libraries, which expressed EGFP at higher intensity compared to the replication-incompetent ZH385, to identify mutations increasing polymerase activities of H13N8 AIVs. By using this approach, several mutations in the polymerase complex that increased polymerase activity in mammalian cells and stabilizing mutations in the HA protein of H5N1 AIVs were identified [21,22]. Additionally, this approach was also carried out to study the high-yield vaccine influenza viruses [28,29].

Genetic mutations provided AIVs opportunities to enhance their replication ability in the new hosts. Previous studies have shown that G228S mutation in HA protein, which presented human receptor binding preference, was identified in H13N8 AIVs [30]. N30D and T215A mutations in the matrix 1 (M1) protein were demonstrated in H13N8 AIVs, which have an increased effect on the pathogenicity of the H5N1 virus in mice [30]. Additionally, a number of mutations increasing the polymerase activity could enhance the replication and virulence of AIVs in mammalian cells and mammalian hosts, such as PB2-I292V, PB2-E627K, and PB2-D701N [31,32,33]. In this study, we screened out the key mutations, PB2-I76T and PB2-I559T, which could enhance the polymerase activity of ZH385 in HEK293T cells through a high-throughput screening system. Position 76 is situated in the PB2 N1 domain (residues 40–110), known for extensive interactions with PB1 [34], while position 559 is located in the PB2 RNA-binding 627-domain (residues 535–693), impacting host adaptation of the influenza virus [35]. Meanwhile, positions 470, 559, 560, and 674 in PB2 protein had previously been reported to contain unique amino acids for seagull-origin H13 AIVs [6]. We also found that PB2-76T and PB2-559T progressively accumulated in H13N8 AIVs and became predominant in 2015. It indicated that H13N8 AIVs were going through the natural selection process.

## 5. Conclusions

In conclusion, H13N8 AIVs circulating in nature have undergone genetic reassortment and mutation events. We evaluated whether the potential reassortment and mutation events could enhance the mammalian adaptation of H13N8 AIVs. Recombination with H9N2 AIVs and genetic mutations could accelerate the evolutionary process of H13N8 AIVs to expand their host ranges. It is necessary to strengthen the surveillance of H13N8 AIVs.

## Figures and Tables

**Figure 1 viruses-16-00329-f001:**
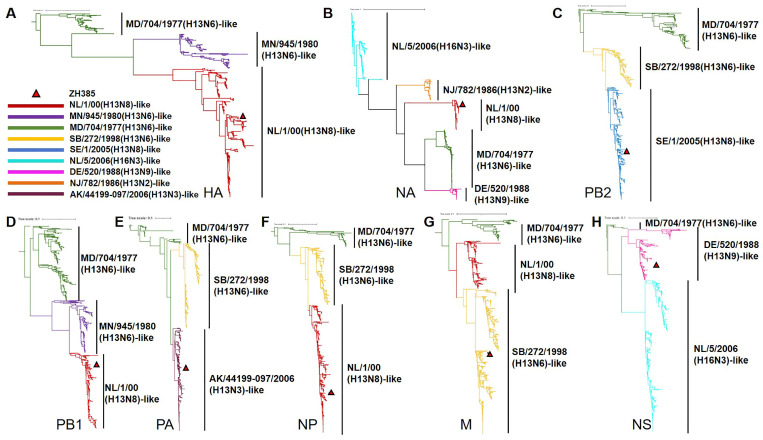
Phylogenetic analysis of H13 AIVs. Phylogenic trees of the (**A**) HA, (**B**) NA, (**C**) PB2, (**D**) PB1, (**E**) PA, (**F**) NP, (**G**) M, and (**H**) NS genes. The phylogenic trees were built using the maximum-likelihood method with rapid bootstrap replicates using FastTree v2.1.11. The nine bars represent the groups of the viruses. The different colors represent different groups. Viral groups and their representatives: NL/1/00(H13N8)-like, A/black-headed gull/Netherlands/1/00(H13N8); MN/945/1980(H13N6)-like, A/gull/Minnesota/945/1980(H13N6); MD/704/1977(H13N6)-like, A/gull/Maryland/704/1977(H13N6); SB/272/1998(H13N6)-like, A/duck/Siberia/272/1998(H13N6); SE/1/2005(H13N8)-like, A/black-headed gull/Sweden/1/2005(H13N8); NL/5/2006(H16N3)-like, A/European herring gull/Netherlands/5/2006(H16N3); DE/520/1988(H13N9)-like, A/ruddy turnstone/Delaware Bay/520/1988(H13N9); NJ/782/1986(H13N2)-like, A/herring gull/NJ/782/1986(H13N2); and AK/44199-097/2006(H13N3)-like, A/glaucous gull/Alaska/44199-097/2006(H13N3). ZH385 is labeled with a triangle in the phylogenic trees.

**Figure 2 viruses-16-00329-f002:**
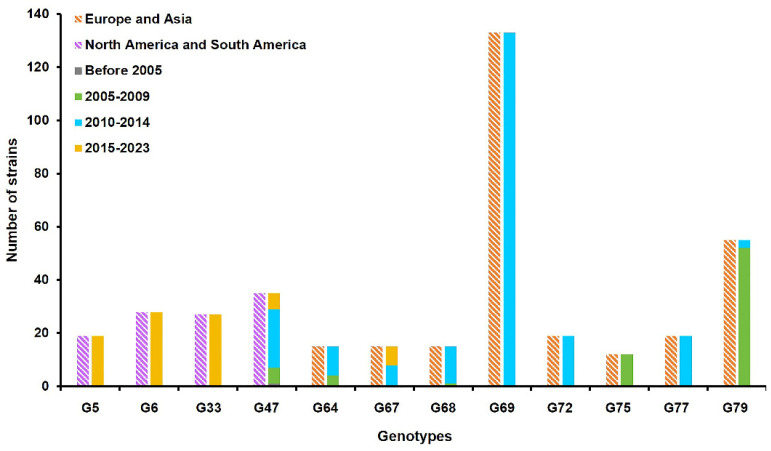
Prevalent genotypes of H13 AIVs. Twelve genotypes possessing more than ten strains were analyzed. The blocks represent the geographic region and collection date of strains. The different colors represent different geographic regions and collection dates.

**Figure 3 viruses-16-00329-f003:**
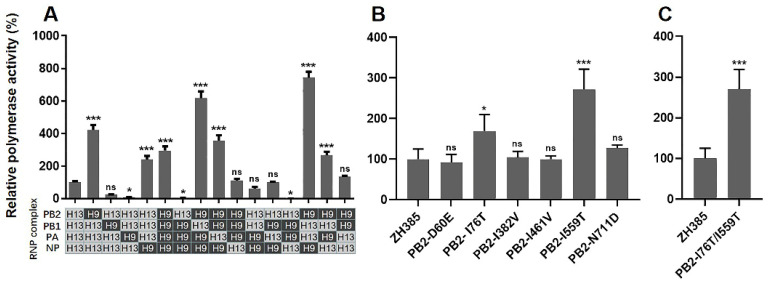
Polymerase activity of recombinant and mutant ZH385 in HEK293T cells. (**A**) Polymerase activity of the 16 RNP combinations between ZH385 (H13N8) and CKLN07 (H9N2). Segments derived from ZH385 and those derived from CKLN07 are denoted by H13 and H9, respectively. (**B**,**C**) Polymerase activity of the ZH385 PB2 mutants. Values shown are means ± SDs from triplicate transfections. The significance of polymerase activity was compared with that of ZH385. Statistical analysis between different groups was performed by using a one-way analysis of variance (ANOVA) test. * (*p* < 0.05), *** (*p* < 0.001), and ns (*p* > 0.05).

**Figure 4 viruses-16-00329-f004:**
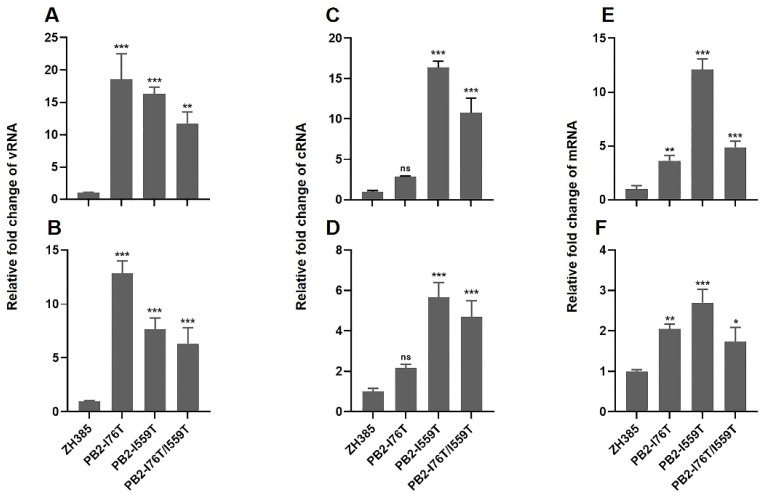
Replication and transcription of the ZH385 PB2 mutants in MDCK cells. (**A**,**B**) The levels of vRNA, (**C**,**D**) cRNA, and (**E**,**F**) mRNA of the ZH385 PB2 mutants. The levels of RNA were determined by qRT-PCR in MDCK cells that were infected for 6 h (**A**,**C**,**E**) and 8 h (**B**,**D**,**F**) with the ZH385 PB2 mutants and normalized to the GAPDH level. The values were standardized to the ZH385, and the values shown are means ± SDs from triplicate infections. Statistical analysis between different groups was performed by using a one-way analysis of variance (ANOVA) test. * (*p* < 0.05), ** (*p* < 0.01), *** (*p* < 0.001), and ns (*p* > 0.05).

**Figure 5 viruses-16-00329-f005:**
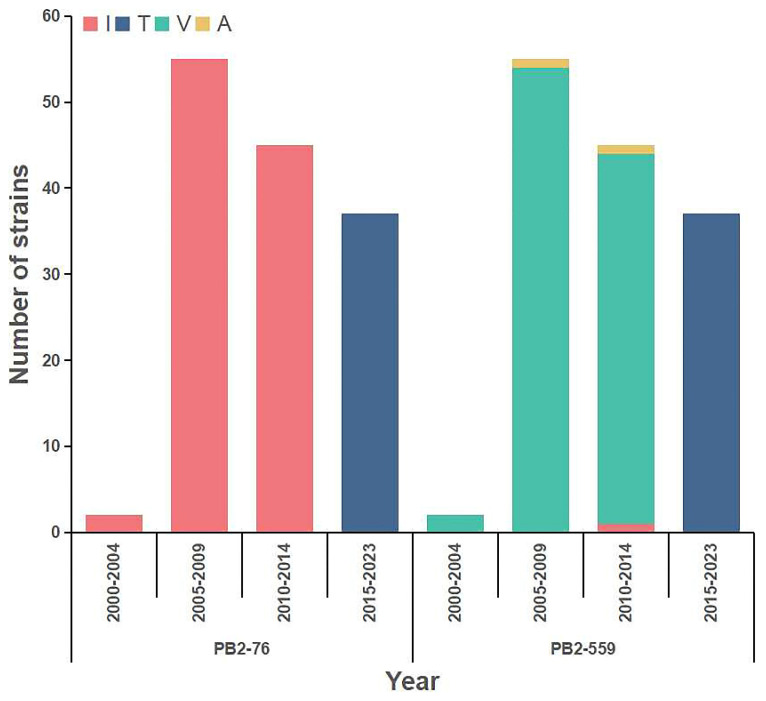
Prevalence of the amino acids at positions 76 and 559 in PB2 proteins of H13N8 AIVs isolated since 2000.

## Data Availability

Data will be made available on request.

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
