# Peer review of "Evolutionary Events Promoted Polymerase Activity of H13N8 Avian Influenza Virus"

_viruses, 2024, doi:10.3390/v16030329_

Round 1
Reviewer 1 Report (Previous Reviewer 2)
Comments and Suggestions for Authors
Comments and Suggestions you can find in the attached file.

Author Response
Please see the attachment.

Reviewer 2 Report (Previous Reviewer 1)
Comments and Suggestions for Authors
This version of the paper prepared by Dr. Meng and his colleagues is significantly improved compared with the previous version. However, I have some major issues, which can be addressed before publication to increase the paper impact.
Authors described a comprehensive phylogenetic analysis of H13Nx viruses including HA, NA and the six internal segments. Authors identified 46 different genotypes for H13Nx viruses, but they excluded HA and NA segments from the genotype analysis. Now, viruses with different NA subtypes belong to the same genotype (e.g., G17 has viruses with N6 and N8; G12 has viruses with N6 and N9, etc), which is incorrect. Authors should revise genotype analysis to include HA and NA lineages for each genotype.
Authors described the “Convergent Analysis of H13 AIV” but the analysis is described poorly, and it does not add any value to the paper in my opinion. I suggest authors to drop this analysis from the paper.
Authors measured the activity of the influenza polymerase complex by co-transfecting the PB2, PB1, PA and NP with a luciferase reporter. They found that reporter activity is higher when the PB2 or NP of H13N8 virus are replaced with PB2 or NP of H9N2. A major limitation of this assay is that the result can be due to not only to differences in the activity of H13N8 and H9N2 proteins but also differences in expression levels (e.g. PB2 of H13N9 expression is lower than expression of PB2 of H9N2). Authors should acknowledge this limitation of their results. Authors described in the main text fold changes between reporter activities but the graphs in Figure 3 show relative luciferase activity. Can authors change the figure 3 to plot fold changes instead of relative luciferase activity.
Authors used replication incompetent ZH385 (H13N8) viruses, which has the HA coding region replaced with eGFP reporter, to measure viral polymerase activity in infected cells, which is an elegant and safe approach to perform this type of analysis. These viruses replicate only in MDCK cells stably expressing HA gene in trans. However, authors do not describe precisely how they prepared and validated the MDCK-HA expressing H13 of ZH385 virus. Can they provide more details about the DNA plasmid used to prepare this cell lines and how they characterized the MDCK cell line stably expressing H13 HA? Moreover, authors need to describe the primers and they validated their assay to measure viral, complementary and messanger RNAs for strand-specific real-time RT-PCR method used to generate results in Figure 4.
It will be helpful to add the PB2 sequence alignment used to prepare the analysis described in Figure 5 or described clearly how the PB2 amino acid numbering is performed.
Comments on the Quality of English LanguageNo comments.
Author Response
Please see the attachment.

Reviewer 3 Report (New Reviewer)
Comments and Suggestions for Authors
In the manuscript “Evolutionary Events Promoted Polymerase Activity of H13N8 Avian Influenza Virus”, the authors analyzes the genetic alterations observed in the evolutionary trajectory of H13 AIVs, highlighting specific natural mutations that could facilitate their adaptation to mammals. The findings presented herein are crucial for comprehending the dynamics of avian influenza originating from wild birds and the associated risks of transitioning to new host species. however some comments that the authors may need to consider to improve the manuscript.
1. Although this manuscript was well written, the language should be slightly improved, and revised by a native English scientist.
2. In line 45-47, the authors describe that H13 AIVs spread to domestic poultry, please provide the corresponding citations.
3. Please describe the characteristics of MDCK -HA cell line.
4. Please provide accession numbers for the genome sequences of A/Eurasian Curlew/Liaoning/ZH-385/2014(H13N8) (ZH385) and A/chicken/China/07/2016(H9N2) (CKLN07), and add more details about CKLN07.
5. Please add more details about qRT-PCR in 2.7 of Materials and Methods
6. In line 176-179, the genotypes of H12 were divided by internal genes which was different with conventional classification methods. Please fully explain.
7. In Table S4, “the phenotype of PB2-T661A” lacks a citation.
Comments on the Quality of English LanguageAlthough this manuscript was well written, the language should be slightly improved, and revised by a native English scientist.
Round 2
Reviewer 2 Report (Previous Reviewer 1)
Comments and Suggestions for Authors
The authors addressed most of the issues I raised in my previous review. I consider that authors should acknowledge as a limitation of their study that the results described in Figure 3 can be due not only to differences in the specific activities of PB2 or NP genes but also to differences in the expression levels in 293 cells (PB2 and NP of H9N2 virus have higher expression in 293 cells than PB2/NP of H13N8 virus).
Comments on the Quality of English LanguageNo comments
Author Response
Please see the attachment.

This manuscript is a resubmission of an earlier submission. The following is a list of the peer review reports and author responses from that submission.
Round 1
Reviewer 1 Report
Comments and Suggestions for Authors
Authors did not check the expression levels for each component of the polymerase complex. Reported differences in luciferase activity can be due to differences in protein expression or in polymerase-specific activity. Some of the studied mutations can also change the expression levels of the protein but they may not change the specific activity. Authors claim that the observed differences in reporter activity are due to changes in the enzymatic activity which is not correct. It will be useful to add comprehensive phylogenetic analyses of the PB2,PB1, PA, NP influenza genes of H13Nx and H9N2 viruses. The paper can be significantly stronger if authors use influenza reverse genetics to rescue these reassortant viruses but this experiment may require BSL3 laboratory.
Reviewer 2 Report
Comments and Suggestions for Authors
Comments are in file attached.

Reviewer 3 Report
Comments and Suggestions for Authors
The interpretation of the polymerase activity test results of H13N8 seems to be incorrect. (Maybe, PB2 451 and 661 mutants with increased polymerase activity were excluded, and NP 350 mutants with decreased polymerase activity were included.) Because of this, Fig 2B, result, discussion, and conclusion also need to be revised. Overall, the explanation of the manuscript is insufficient and there is a lack of organic logic between sentences or paragraphs.
Comments on the Quality of English LanguageEnglish correction by a native speaker with expert knowledge is required.
Reviewer 4 Report
Comments and Suggestions for Authors
The authors investigated by use of a dual luciferase reporter minigenome assay the effects on a H13N8 backbone (isolate from 2014) reassorted with RNPs of H9N2. PB2 and NP of H9 enhanced the activity. In addition, two mutations in PB2 and 1 in NP that emerged in recent H13 viruses enhanced polymerase activity of the 2014 isolate. The authors conclude that there are increasing zoonotic risks of H13 viruses.
This is a problematic manuscript:
1. The authors have changed the genome of an influenza to enhance polymerase activity. They claim, these mechanisms could increase zoonotic risks. In summary, this would fit the definition of gain-of-function research.
2. The authors base their study design on the assumption that there are options for a reassortment between H13 and H9N2 viruses in poultry in China. However, they do not provide references indicating that H13 viruses have ever been detected in poultry (lines 45-6) although they cite data on H13 seroconversion in chickens. This is insufficient data.
3. They have analysed sequence of H13 viruses obtained since 2000 for coding mutations in polymerase RNPs. However, they do not lay open from which regions these viruses stem. The changes they observed could be merely related to a bias in geographic origin of these viruses (i.e., American vs Eurasian).
4. There are many inconsistencies in the text and linguistic problems. Examples: L 37: What is meant by “survive for several days”? L41 What is the differences between inter and transcontinental? L172: …mutations can significantly may increase… If it is significant there is no “can” or “may”; significance values should be shown. L210-3: This is purely speculative.
5. H13 genes have already reassorted into a Eurasian lineage of the HPAIV H5N1 (genotype BB). However, the authors do not mention this anywhere in their manuscript.
All in all, I regret not to be able to recommend this manuscript for publication.
Comments on the Quality of English LanguageSee the part above.
Round 2
Reviewer 2 Report
Comments and Suggestions for Authors
Dear authors,
This version of the article is quite well. Supplemented material is enough. It helps to understand the work. But I have still some notes.
1. Figure 1. The names of key strains are too small to read. If possible, it is better to enlarge their font. At least this needs to be done for the root strains for each gene.
2. Figure 3A.
It seems you have confused the designation order of proteins (genes) PB2, PB1, PA, and NP in Figure 3A. Since description in the text (lines 256-260) does not coincide to this figure. The similar Figure 1 from the first version of the article was correct.
3. Subsection 3.7.
Lines 353-354. The results suggested that PB2-76T and PB2-559T might be obtained as the species barrier crossing keys during the evolution of H13N8 AIVs.
This conclusion is unproved, since you have not pointed species of virus host which were sampled for analysis. All tested viruses were obviously belonging to viruses which were isolated from birds. But what kind of species were that birds? You didn't provide that information. Therefore, it had be better to remove the last sentence.
See, please, applied file.

Reviewer 3 Report
Comments and Suggestions for Authors
The authors added content that differs significantly from the old version. However, the authors submitted manuscripts with recorded revisions, making it very difficult for me to review them. Because these are PDF files, they are difficult to adjust. It seems that the authors will have to resubmit or remove the recorded changes.